# Enhanced Sample Self-Revised Network for Cross-Dataset Facial Expression Recognition

**DOI:** 10.3390/e24101475

**Published:** 2022-10-17

**Authors:** Xiaolin Xu, Yuan Zong, Cheng Lu, Xingxun Jiang

**Affiliations:** 1Key Laboratory of Child Development and Learning Science of Ministry of Education, Southeast University, Nanjing 210096, China; 2School of Biological Science and Medical Engineering, Southeast University, Nanjing 210096, China

**Keywords:** cross-dataset facial expression recognition, facial expression recognition, unsupervised domain adaptation, transfer learning

## Abstract

Recently, cross-dataset facial expression recognition (FER) has obtained wide attention from researchers. Thanks to the emergence of large-scale facial expression datasets, cross-dataset FER has made great progress. Nevertheless, facial images in large-scale datasets with low quality, subjective annotation, severe occlusion, and rare subject identity can lead to the existence of outlier samples in facial expression datasets. These outlier samples are usually far from the clustering center of the dataset in the feature space, thus resulting in considerable differences in feature distribution, which severely restricts the performance of most cross-dataset facial expression recognition methods. To eliminate the influence of outlier samples on cross-dataset FER, we propose the enhanced sample self-revised network (ESSRN) with a novel outlier-handling mechanism, whose aim is first to seek these outlier samples and then suppress them in dealing with cross-dataset FER. To evaluate the proposed ESSRN, we conduct extensive cross-dataset experiments across RAF-DB, JAFFE, CK+, and FER2013 datasets. Experimental results demonstrate that the proposed outlier-handling mechanism can reduce the negative impact of outlier samples on cross-dataset FER effectively and our ESSRN outperforms classic deep unsupervised domain adaptation (UDA) methods and the recent state-of-the-art cross-dataset FER results.

## 1. Introduction

Facial expressions reveal human emotional conditions and mental outlook visually. Therefore, automatic facial expression recognition (FER) has great application prospects in the fields of human–computer interface (HCI) [1] and medical care [2]. Over the last decade, increasing researchers have paid attention to the research of facial expression recognition. Several effective methods [3,4,5,6] have driven the progress of FER. At the same time, the emergence of a variety of facial expression datasets [7,8,9,10,11] has also greatly improved FER performance.

Although FER has made great strides in recent years, it should be pointed out that most of the FER methods mentioned above are designed and evaluated under an ideal assumption that the training and testing of facial expression images come from the same dataset. Under this assumption, the training and testing samples follow the same or similar distribution. However, in practical scenarios, the training and testing samples usually belong to different datasets, which means the training and testing samples may have a difference in the pose, illumination, background, and level of expressiveness [12]. In this case, the performance of most aforementioned FER methods may drop sharply due to the feature distribution mismatch existing between the training and testing sets, thus bringing researchers a greater challenge (i.e., cross-dataset FER) compared with the traditional FER.

However, due to the influence of the annotation environment and cultural background, the annotators of the FER dataset may produce extreme subjective annotation deviation [12]. In addition, several other interference factors, such as sample quality (e.g., occlusion and sample clarity) and racial bias [13], may also produce some outliers in facial expression datasets as shown in Figure 1. Wang et al. [14] declared that outlier samples in facial expression datasets could be harmful to the model for learning useful facial expression features. Xu et al. [15] also revealed the performance degradation of other cross-dataset tasks, through extensive experiments, caused by learning incorrect outlier samples. Similarly, in dealing with cross-dataset FER, these outlier samples can also inevitably break the modeling of the relationship between the source domain and its corresponding label information. Therefore, the unsupervised domain adaptation (UDA) models could fail to learn the discriminative facial expression features. In this case, it is hard for the UDA models to cope with the cross-database FER tasks, although they successfully eliminate the feature distribution difference between the source and target domains with well-designed strategies. Therefore, it is crucial to consider the outlier samples in dealing with cross-dataset FER tasks.

To deal with this problem, we proposed an effective network termed as sample self-revised network (SSRN) in our conference work of [15]. SSRN is committed to learning a feature extractor that is more robust to both source and target domains. By reducing the influence of outlier samples in the source domain on model training and aligning the target domain with the revised source domain at the feature level, SSRN can obtain the domain-invariant features for both source and target domains. Precisely, the proposed SSRN consists of three essential modules: outlier perception module, outlier perception coefficient (OPC) revision module, and feature transfer module. Given a batch of samples from source and target domains, respectively, the CNN backbone can be used to extract facial features. Then, the outlier perception and the OPC revision modules are used to dynamically perceive outlier samples in the source domain and mitigate the influence of these outliers on the model training. After that, the well-designed feature transfer module enforces the outlier samples to share the same feature distribution with other samples in the source domain so as to revise these outlier samples. In addition, features of source and target domains can also be aligned to learn more robust domain invariant features with the feature transfer module.

In this paper, we strengthen our conference work on SSRN [15], and further propose the enhanced sample self-revised network (ESSRN). Specifically, we modify our original SSRN from two aspects. First, the class imbalance of the FER dataset can sharply degrade the domain adaptation performance [16]. Inspired by [17], we add conditional maximum mean discrepancy (MMD) [18] while dealing with outliers revision and domain adaptation in the feature transfer module. In addition to aligning the marginal distribution considered in previous works, we further align the class-conditional distribution by narrowing the feature distance of each category between the source and the target domains. Second, we added a feature reconstruction layer in the revision process. The feature reconstruction layer performs linear reconstruction of outlier samples at the feature level so that the revised samples can mitigate the inconsistency of modeling the relationship between the source domain and their label information. In summary, besides the original contributions in our preliminary work [15], in this paper, we further improve the outliers revision function and the unsupervised domain adaptation process to learn better domain-invariant features to describe facial expressions. Therefore, we achieve more promising performance in dealing with cross-dataset FER.

The rest of this paper is organized as follows. We review recent works about cross-dataset facial expression recognition and methods for outliers in Section 2. In Section 3, we introduce our proposed ESSRN in detail. Section 4 presents four public facial expression datasets and our experiment protocol. Extensive experiments and evaluations are provided in Section 5. Finally, the work of this paper is concluded in Section 6.

## 2. Related Works

### 2.1. Cross-Dataset Facial Expression Recognition

To deal with cross-dataset FER, current mainstream approaches treat it as an unsupervised domain adaptation (UDA) problem and design corresponding methods [12,16,19,20,21,22,23,24,25]. For instance, Yan et al. [19] proposed an unsupervised domain adaptive dictionary learning method (UDADL). Combining dictionary learning with unsupervised domain adaptation, UDADL achieved satisfactory running time and recognition accuracy results. Zheng et al. [20] proposed a transductive transfer subspace learning method (TTRLSR). Combining a labeled image set from the source domain with an unlabeled auxiliary image set from the target domain, TTRLSR jointly learns a discriminative subspace to deal with cross-dataset FER. In the work of [21,22], Zong et al. aimed to learn a domain regenerator that regenerates source samples (target samples) sharing the same or similar feature distribution as the target samples (source samples). Other approaches based on deep learning have also achieved promising results on cross-dataset FER. Wang et al. [23] utilized additional data generated by the generative adversarial network (GAN) to optimize the cross-dataset performance of FER. Zhou et al. [24] proposed an uncertainty-aware cross-dataset facial expression transfer network (UA-ETN). Aligning the marginal and class-conditional distribution, UA-ETN can enhance the generalization ability of cross-dataset FER. In the work of [16], Li et al. revealed the influence of class imbalance on cross-dataset FER and proposed a re-weighting method termed as deep emo-transfer network (DETN), which utilizes re-weighted maximum mean discrepancy (MMD) [18] to improve the generalization performance of the model. Based on the above work, Li et al. explored the bias in FER datasets in depth in the work of [12]. They pointed out the influence of inconsistent conditional probability distributions in source and target domains on cross-dataset FER. To this end, they proposed emotion-conditional adaption network (ECAN), which considers both marginal and conditional distributions in cross-dataset FER. Xie et al. [25] proposed an adversarial graph representation adaptation (AGRA) that combines graph representation propagation with adversarial learning to learn local and global joint features to realize cross-dataset FER better.

### 2.2. Methods for Outliers

Many solutions have been proposed to deal with outlier samples [14,26,27,28] existing in all kinds of datasets. Among these datasets, the noisy samples generated by subjective annotation are the most widely studied. Aiming at the problem of inconsistent annotation in dataset annotation, Dawid et al. [26] used the EM algorithm to estimate the quality of each annotator in the crowdsourcing process to determine the labels. This study provided a theoretical basis for the later calibration of database labels. Li et al. [28] proposed a unified distillation framework, which uses information from a small clean dataset and label relations in a knowledge graph, to avoid learning from noisy labels. In contrast, other approaches refuse to employ a clean set of data. Ref. [29] assumed additional constraints or distributions on the noise labels to learn the relationship between the noisy samples and the corresponding latent labels.

Recently, outlier problems have been focused on conventional FER tasks, and numerous methods have been proposed to handle outlier samples. For example, as the first work to consider outlier samples in FER, Zeng et al. [27] indicated the outlier samples were caused by inconsistent annotations among different FER datasets and proposed an inconsistent pseudo annotations to latent truth (IPA2LT) framework. IPA2LT artificially creates outlier samples by attaching multiple pseudo-labels to the samples labeled by humans or the learned models to learn the potential mapping between images and truth. Employing this latent mapping, the model can obtain better performance from multiple datasets with inconsistent annotations. Recently, Wang et al. [14] considered the outlier samples produced by the incorrect labeling and raised the Self-Cure Network (SCN) to cope with FER. The basic idea of SCN is to suppress these outlier samples in the model learning by seeking and relabeling them from the training facial expression samples.

## 3. Methods

Outliers can seriously damage the extraction of domain invariant features on cross-dataset FER. To address these issues, we propose a simple yet effective method called enhanced sample self-revised network (ESSRN). The description of ESSRN in this section is arranged as follows. Firstly, the overview of the proposed ESSRN is presented in Section 3.1. Then we review the previous work of SSRN [15]. Three important components of SSRN are introduced in detail in Section 3.2, Section 3.3 and Section 3.4, respectively. Based on SSRN, we are able to propose the ESSRN in this paper. Therefore, after that, we demonstrate the improvements of ESSRN over SSRN in Section 3.5. At last, the specific optimization process of ESSRN is given in Section 3.6.

### 3.1. Overview of ESSRN

In this part, we introduce the overview of ESSRN. ESSRN consists of three crucial modules: outlier perception, outlier perception coefficient (OPC) revision, and feature transfer, which are shown in Figure 2. These modules work together to implement the proposed novel outliers handling mechanism, whose aim is to seek outlier samples and then suppress them in dealing with cross-dataset FER. Firstly, the outlier perception module seeks outlier samples dynamically and suppresses the model from learning incorrect knowledge from outlier samples. Secondly, the OPC revision module includes a constraint to strengthen the outliers detection ability of the outlier perception module. Finally, the feature transfer module can obtain more robust domain invariant features in dealing with cross-dataset FER by aligning the marginal and class-conditional distribution of the revised source and target domains. Detailed implementation will be described in the following part of this section. Before that, several necessary notations are introduced in the following, including source features and target features, which are denoted by $F^{s}=\left[f_{1}^{s},f_{2}^{s},\cdots ,f_{N_{s}}^{s}\right]\in R^{D\times N_{s}}$ and $F^{t}=\left[f_{1}^{t},f_{2}^{t},\cdots ,f_{N_{t}}^{t}\right]\in R^{D\times N_{t}}$. $N_{s}$ and $N_{t}$ are the number of source and target datasets, and *D* represents the dimension of feature vectors of source and target samples.

### 3.2. Outlier Perception

Outlier perception perceives the level of outlier samples in the source domain and allows the SSRN to apply attention to different data during the model training adaptatively. It is generally recognized that facial images with low resolution or serious occlusion will be given a high level, and unambiguous images will be assigned to a low level. To realize the outlier perception, Outlier Perception employs a fully connected layer and a sigmoid activation function which takes the feature $F^{s}$ as input and considers the output scalar $\alpha_{i}\left(i\in 1,2,\cdots ,N\right)$ as the outlier perception coefficient (OPC) of each image. Its implementation is formulated as
(1)$$\alpha_{i}=\sigma \left(w_{op}^{T}f_{i}^{s}\right),$$
where $\sigma \left(\cdot \right)$ represents the sigmoid activation function, and $w_{op}$ represents parameters of the fully connected layer. The output α of the above formula is inversely proportional to the outlier level of each sample, which means the higher the image quality or, the more consistent the annotation is, the higher α will be assigned; conversely, the lower the image quality is, the lower corresponding α will be.

In order to allow the model to learn dynamically from the samples at different weights, a suitable loss function [30], i.e., outlier perception loss, is chosen to adjust the contribution of source data with different OPC. Outlier perception loss (OP-Loss) can be formulated as
(2)$$q_{ij}=\frac{e^{\alpha_{i}e_{j}^{T}W_{cls}^{T}f_{i}^{s}}}{\sum_{k=1}^{C}e^{\alpha_{i}e_{k}^{T}W_{cls}^{T}f_{i}^{s}}},$$
(3)$$L_{OP}=-\frac{1}{N}\sum_{i=1}^{N}\sum_{j=1}^{C}p_{ij}\log q_{ij},$$
where $p_{ij}$ and $q_{ij}$ represent the ground truth and the prediction of the *j*-th class of the *i*-th sample, respectively, $e_{j}$ is a one-hot vector, whose *j*-th element is one and the others are equal to zero. $W_{cls}$ represents parameters of the classifier. By employing the weighted cross-entropy loss function, the model can reduce the interference of outlier samples to the model training and pay more attention to the samples with low outlier levels.

### 3.3. OPC Revision

The aforementioned outlier perception module can perceive the level of outlier samples in the source domain. To further guarantee that the meaningful mapping relationship between facial expression features and outlier level can be learned correctly, the OPC revision module is proposed to revise and constrain the α generated from the outlier perception module. The module first takes the OPC of each sample in the source domain as input and then arranges them in descending order. After that, it divides them into the high-coefficient and low-coefficient groups by a certain ratio β, among which the samples in the low-coefficient group can be regarded as the outlier samples we mentioned above. In addition, a newly designed loss function, i.e., OPC revision loss (OR-Loss), is proposed to ensure that the low-coefficient and high-coefficient groups can be separated. The formulation follows can guarantee that the mean value of the high-coefficient group is higher than the low-coefficient group with a certain margin,
(4)$$L_{OR}=\max\left\{0,\mathrm{Margin}-\left(\alpha_{H}-\alpha_{L}\right)\right\},$$
where Margin represents the set margin between the mean value of the high-coefficient and low-coefficient groups. $\alpha_{H}$ and $\alpha_{L}$ represent mean values of the high-coefficient and low-coefficient groups calculated by
(5)$$\alpha_{H}=\frac{1}{M}\sum_{i=1}^{M}\alpha_{i}$$
and
(6)$$\alpha_{L}=\frac{1}{N_{s}-M}\sum_{i=M+1}^{N_{s}}\alpha_{i},$$
where the number of samples in high-coefficient and low-coefficient groups is $M=N_{s}\times \beta$ and $N_{s}-M$, respectively.

### 3.4. Feature Transfer

With the help of the outlier perception module and OPC revision module, the model can learn the OPC of each sample relative to the sample feature center in the source domain correctly. However, outlier samples could make the model learn the incorrect facial expression features and deter the model from aligning the features in the source and target domains. Therefore, the feature transfer module is proposed to mitigate these problems.

Before introducing the implementation of the feature transfer module in detail, let us introduce some necessary theoretical bases.

#### 3.4.1. Preliminary

As the most widely used loss function in domain adaptation, maximum mean discrepancy (MMD) [18] is mainly applied for comparing distributions between two domains [31], which can be formulated as
(7)$${\mathrm{MMD}}^{2}\left(X,Y\right)={∥\frac{1}{n}\sum_{i=1}^{n}\phi \left(x_{i}\right)-\frac{1}{m}\sum_{j=1}^{m}\phi \left(y_{j}\right)∥}_{H}^{2},$$
where $x_{i}$ and $y_{j}$ represent the sample from the domain *X* and *Y*. *n* and *m* represent the number of samples in the domain *X* and *Y*. H denotes the reproducing kernel Hilbert space (RKHS). Projecting the data into H by the mapping function $\phi \left(\cdot \right)$, we can transform the inner product of function in the RKHS to the form of kernel function:(8)$$K\left(x,y\right)={\langle \phi \left(x\right),\phi \left(y\right)\rangle }_{H},$$
where $K\left(x,y\right)$ represents the kernel function. In most UDA tasks, the most commonly used kernel function is the Gaussian kernel function:(9)$$K\left(x,y\right)=e^{\frac{{-∥x-y∥}^{2}}{2\sigma^{2}}},$$
which can map the data to the infinite-dimensional space. The unbiased estimation expression of MMD after expanding the square of the (Equation 7) can be formulated as follows:(10)$$\begin{array}{cc} {\mathrm{MMD}}^{2}\left(X,Y\right) & =\frac{1}{n^{2}}\sum_{i=1}^{n}\sum_{i^{\prime }=1}^{n}K\left(x_{i},x_{i^{\prime }}\right)\\ \; & +\frac{1}{m^{2}}\sum_{j=1}^{m}\sum_{j^{\prime }=1}^{m}K\left(y_{j},y_{j^{\prime }}\right)\\ \; & -\frac{2}{nm}\sum_{i=1}^{n}\sum_{j=1}^{m}K\left(x_{i},y_{j}\right). \end{array}$$

In addition, we further replace MMD with a single fixed kernel by multi-kernel MMD (MK-MMD) [18], which can be formulated as follows:(11)$$K:=\left\{K=\sum_{u=1}^{d}\beta_{u}K_{u}:\beta_{u}\geq 0,\forall u\in \left\{1,\ldots ,d\right\}\right\}.$$

MK-MMD obtains the optimal kernel by linearly weighting multiple kernels ${\left\{k_{u}\right\}}_{u=1}^{d}$ with weight $\beta_{u}$, which is more powerful in representation compared with the single-kernel MMD.

#### 3.4.2. Feature Transfer Module

The feature transfer module embeds MK-MMD in the task-specific feature layer of the deep network, which can be expressed as follows:(12)$$L_{MMD}=\mathrm{MMD}\left(F_{H}^{s},F_{L}^{s}\right)+\mathrm{MMD}\left(F^{s},F^{t}\right),$$
where $F_{H}^{s}$ and $F_{L}^{s}$ represent the feature vectors of the high-coefficient and low-coefficient groups in the source domain. The loss function of the feature transfer module is composed of two parts. The first half of (Equation 12) reduces the feature distribution distance of high-coefficient group and low-coefficient group samples in the source domain, which revises outlier samples detected by the outlier perception module. The second half of (Equation 12) further ensures that the target domain features can be aligned with the revised source domain features so that the model can learn more robust domain invariant features.

### 3.5. From SSRN to ESSRN

Through the above content, we recalled the specific implementation details of SSRN. In this article, we improved SSRN based on previous work and further propose ESSRN. To be more specific, in the implementation of ESSRN, we not only paid attention to the marginal distribution difference between the source and target domains, but also minimized the conditional distribution difference between the corresponding categories of the source and target domains. We retained the structure of outlier perception and OPC revision modules in the original SSRN, and mainly improved the feature transfer module. In the feature transfer module of ESSRN, we newly proposed two functional structures: feature revision layer and domain adaptation layer. With the help of the feature transfer module, the feature transfer module could first better revise the outlier samples in the source domain at the feature level so that the revised source domain could keep a relatively consistent distribution. Then, the domain adaptation layer further narrowed the distribution difference between the revised source and target domains to extract robust domain invariant features.

#### 3.5.1. Feature Revision Layer

In order to further reduce the influence of outlier samples on domain invariant feature learning, we add an additional feature revision layer in the feature transfer module compared with the previous work. The feature revision layer first employs a fully connected layer to revise the outliers in the source domain linearly. Then, we propose a loss function based on MK-MMD to force the generated feature to share the same or similar feature distribution with the source domain. The proposed loss function jointly minimizes the MMD distance of marginal and conditional probability distributions between outliers and the source domain to better revise the outliers at the feature level. The entire feature revision layer can be formulated as follows:(13)$$F_{L}^{s+}=w_{FR}^{T}F_{L}^{s},$$
(14)$$\begin{array}{cc} L_{MMD1} & ={∥\frac{1}{M}\sum_{i=1}^{M}\phi \left(f_{H}^{s_{i}}\right)-\frac{1}{N_{s}-M}\sum_{i=1}^{N_{s}-M}\phi \left(f_{L}^{s_{i}+}\right)∥}_{H}^{2}\\ \; & +\frac{1}{C}\sum_{j=1}^{C}{∥\frac{1}{M_{j}}\sum_{i=1}^{M_{j}}\phi \left(f_{H}^{s_{ij}}\right)-\frac{1}{N_{s_{j}}-M_{j}}\sum_{i=1}^{N_{s_{j}}-M_{j}}\phi \left(f_{L}^{s_{ij}+}\right)∥}_{H}^{2}, \end{array}$$
where $F_{L}^{s+}$ represents the revised outlier feature, and $w_{FR}$ is the parameters of the fully connected layer for feature revision. $f_{H}^{s_{ij}}$ and $f_{L}^{s_{ij}+}$ denote the *i*th feature with the *j*th class from $F_{H}^{s}$ and $F_{L}^{s+}$ respectively. $M_{j}$ and $N_{s_{j}}$ are the corresponding sample numbers satisfying $\sum_{j=1}^{C}M_{j}=M$ and $\sum_{j=1}^{C}N_{s_{j}}=N_{s}$.

#### 3.5.2. Domain Adaptation Layer

In the previous work of [15], we embedded MK-MMD in the task-specific feature layer of the feature transfer module, which only aligns the source and target domains’ marginal distribution and ignores the distribution difference between source and target domains in each category. Inspired by the success of the work of [17], we propose a more adaptable loss function in the domain adaptation layer. Thus, we can simultaneously narrow the difference between the marginal and class-conditional distribution between source and target domains. The newly proposed loss function can be expressed as
(15)$$\begin{array}{cc} L_{MMD2} & ={∥\frac{1}{N_{s}}\sum_{i=1}^{N_{s}}\phi \left(f_{i}^{s}\right)-\frac{1}{N_{t}}\sum_{i=1}^{N_{t}}\phi \left(f_{i}^{t}\right)∥}_{H}^{2}\\ \; & +\frac{1}{C}\sum_{j=1}^{C}{∥\frac{1}{N_{s_{j}}}\sum_{i=1}^{N_{s_{j}}}\phi \left(f_{ij}^{s}\right)-\frac{1}{N_{t_{j}}}\sum_{i=1}^{N_{t_{j}}}\phi \left(f_{ij}^{t}\right)∥}_{H}^{2}, \end{array}$$
where $f_{ij}^{s}$ and $f_{ij}^{t}$ denote the *i*th feature with the *j*th class from $F^{s}$ and $F^{t}$, respectively. $N_{s_{j}}$ and $N_{t_{j}}$ are the corresponding sample numbers satisfying $\sum_{j=1}^{C}N_{s_{j}}=N_{s}$ and $\sum_{j=1}^{C}N_{t_{j}}=N_{t}$.

We accumulate (Equation 14) and (Equation 15) as the loss function of the feature transfer module, i.e., MMD loss, which can be formulated as follows:(16)$$L_{MMD}=L_{MMD1}+L_{MMD2}.$$

The final formulation of the proposed ESSRN is accumulated by (Equation 3), (Equation 4) and (Equation 7), which can be written as
(17)$$L=L_{OP}+\lambda L_{MMD}+\gamma L_{OR},$$
where λ and γ are the trade-off factors to balance the proposed modules.

### 3.6. Optimization of ESSRN

In this section, how the network is optimized is explained in detail. In order to calculate the $L_{MMD}$, we need to give the pseudo label of the target domain. Therefore, we use the alternated direction method to optimize the whole ESSRN.

Specifically, the model optimization process is presented in Algorithm 1. We first randomly initialize the parameters of ESSRN, i.e., $w_{op}$, $w_{cls}$ and $w_{FR}$, and then predict the pseudo labels $L_{p}^{t}$ of the target domain. Subsequently, we perform the following two major steps until the total loss L is equal to zero or the number of iterations reaches the maximal iteration $N_{iter}$:1.Calculating the total loss L according to $L_{p}^{t}$, and updating the network parameters θ by the optimization algorithm, e.g., SGD and Adam.2.Fix parameters θ, and update the pseudo target domain labels $L_{p}^{t}$.
**Algorithm 1** The detailed procedures for optimization of ESSRN**Input:** Source Image Features: $F^{s}=\left[f_{1}^{s},f_{2}^{s},\cdots ,f_{N_{s}}^{s}\right]$,
  Target Image Features: $F^{t}=\left[f_{1}^{t},f_{2}^{t},\cdots ,f_{N_{t}}^{t}\right]$,
  Trade-off Parameters: λ and γ,
  Learning Rate: lr,
  Maximal Iterations: $N_{iter}$
**Output:** Optimal Network Parameters: $\theta =\left\{w_{op},w_{cls},w_{FR}\right\}$ and Target Labels: $L_{p}^{t}$
1:Initialize the network parameters: θ, total loss: $L=L_{OP}+\lambda L_{MMD}+\gamma L_{OR}$, and iteration indicator: iter=02:**while**L≠0 and $iter<N_{iter}$
**do**3:   iter=iter+1;4:   Fix the network parameters: θ, Predict the target domain pseudo label $L_{p}^{t}$;5:   Fix the target domain pseudo label $L_{p}^{t}$, calculate L;6:   Update the network parameters: θ:7:   $\;\nabla_{\theta }\leftarrow \frac{\partial (L_{OP}+\lambda L_{MMD}+\gamma L_{OR})}{\partial \theta }$;8:   $\;\theta^{n+1}\leftarrow \theta^{n}-lr*\nabla_{\theta }$;9:**end while**


## 4. Experiment

In this section, we first describe four public facial expression datasets and then presented our experiment protocol. Finally, we demonstrate the implementation details of our cross-dataset FER experiments.

### 4.1. Datasets

Four public available facial expression datasets, including FER2013 [9], RAF-DB [10], CK+ [8], and JAFFE [7], are applied to evaluate the proposed ESSRN.

**FER2013** is a large-scale facial expression dataset consisting of 35,887 facial images of size 48 × 48 pixels. Each image is labeled by one of seven basic expression categories, i.e., angry, disgust, fear, happy, sad, surprise, and neutral. The dataset is further divided into 18,709 training samples, 3589 validation samples, and 3589 testing samples. In the experiments, we adopted all the original facial images.

**RAF-DB** is a real-world facial expression dataset that contains 19,672 facial images collected from the Internet. We use its single-labeled subset consisting of 15,339 samples, which is divided into a training set of 12,271 images and a test set of 3068 images, and each sample was assigned one of seven basic expressions.

**Extended Cohn–Kanade (CK+)** is a lab-controlled dataset that has 123 subjects and records their 593 facial expression video clips. Each video clip is annotated by one of seven expressions, including angry, disgust, fear, happy, sad, contempt, and surprise. In the experiments, to be relatively consistent with other datasets, we only used emotional data for six of these categories, i.e., angry, disgust, fear, happy, sad, and surprise, as well as neutral. We extracted the last three peak frames from the video clips labeled with one of six target categories to serve as expression samples and chose the first frame from these labeled sequences as the neutral expression.

**JAFFE** is also a lab-controlled dataset that contains 213 gray-scale images from 10 Japanese female expressers. Expressers were asked to pose seven facial expressions. All images in the JAFFE are employed in our experiments.

The aforementioned datasets have significant differences in pose, illumination, background, level of expressiveness, and image quality, which can be seen in the typical dataset images in Figure 3. Detailed sample statistics of these datasets are listed in Table 1.

### 4.2. Experiment Protocol

To evaluate the proposed ESSRN, we resized all facial images to 112 × 112 pixels and then designed a total of 12 experiments across the above four datasets in pairs, which are denoted by *R→J, R→C, R→F, J→R, J→C, J→F, C→R, C→J, C→F, F→R, F→J*, and *F→C*, where *R, J, C,* and *F* are the abbreviations of RAF-DB, JAFFE, CK+, and FER2013, and the left and right sides of → represent the source domain and target domain, respectively. As for the performance metric, we employed the recognition accuracy, which is calculated by T/N × 100%, where T is the number of correct predictions and N is the total sample number in the target dataset. In addition, all feature extraction networks in the experiment are with the widely used ResNet-18 [32].

### 4.3. Implementation Details

In this part, we illustrate our experiment implementation in detail, including pre-processing, model training, and hyper-parameters setting stages. In the pre-processing stage, face images are detected and aligned by MTCNN [33] and further transformed to gray-scale. We adjusted the image by resizing the image to 128 × 128 and then randomly cropping to 112 × 112, as well as adding the horizontal flip with the probability of 50%. For the training details, our ESSRN is trained on one Tesla A10 GPU, with the implementation in Pytorch.

The CNN backbone ResNet-18 [32] was pre-trained on ImageNet dataset [34] and the facial features with the dimension of 512 are extracted after the average pooling layer of ResNet-18. We ran our model using stochastic gradient descent (SGD) with an initial learning rate of 0.0001 for the backbone parameters and an initial learning rate of 0.01 for other parts, which are divided by 10 after every 20 epochs, a momentum of 0.9, and a weight decay of 0.0005. For the hyper-parameters setting, the division ratio β was set to 0.7, and the *margin* represents the difference between the mean value of high and low groups was set to 0.15 by default. The ratio of $L_{OP}$, $L_{MMD}$ and $L_{OR}$ will be discussed in the evaluation of trade-off parameters of Section 5. Further, the influence of these three losses will be explored in the ablation study of Section 5.

## 5. Results and Discussion

In this section, we present the experimental results of the proposed ESSRN. We first show the accuracies of our ESSRN across RAF-DB, JAFFE, CK+, and FER2013 in Section 5.1. Then, the sensitivity analysis of the trade-off parameters is discussed in Section 5.2. Finally, the influence of three modules of ESSRN is investigated in Section 5.3.

### 5.1. Results across RAF-DB, JAFFE, CK+, and FER2013

The results of 12 experiments across RAF-DB, JAFFE, CK+, and FER2013 datasets in pairs are shown in this subsection. To offer a relatively fair comparison, we also chose two well-performing deep domain adaption methods, i.e., DANN [35] and DAN [36], to conduct the experiments. In addition, ResNet-18 was chosen as the backbone network for all the comparable experiments to provide a fairer comparison. At last, we also compared our method with recent state-of-the-art cross-dataset FER methods. The experimental results of various UDA methods are presented in Table 2. Several interesting findings can be observed from the experimental results.

Firstly, compared with the classical DAN and DANN methods, our ESSRN outperforms two traditional UDA methods in all experiments. The average recognition accuracy of these three methods shows that ESSRN is 6.95% higher than the DANN method and 2.97% higher than the classic DAN method, which shows the superior performance of our ESSRN in cross-dataset tasks more fairly and intuitively. This is because, compared with traditional UDA methods, especially DAN, our method takes outlier samples in the source domain into consideration and proposes corresponding modules to alleviate its impact on cross-dataset FER tasks.

Secondly, our ESSRN achieved better results in 9 of 12 experiments compared with the recent state-of-the-art cross-dataset FER methods (i.e., DETN [16], ICID [37], ECAN [12], and AGRA [38]), indicating the effectiveness of the proposed outliers handling mechanism in dealing with cross-dataset FER tasks.

Thirdly, compared with the previous SSRN, the proposed ESSRN achieved better results in 9 of the 12 experiments and improved by 2.94% in average accuracy. These results indicated that the newly added feature revision layer and the domain adaptation layer can improve the cross-dataset FER ability by paying extra attention to the difference in class-conditional distribution.

Fourthly, we believe that the source dataset largely determined the final performance in cross-dataset facial expression recognition tasks. Therefore, we calculated the average performance of the 12 experiments above with the same source domains, as shown in Table 3. We compared DAN, which generally has good performance in the classical UDA method, with our ESSRN method and calculated the performance increase based on DAN. According to the experimental results, when the source domains are in-the-wild datasets (e.g., RAF-DB and FER2013), the experimental performance is much higher than that of the lab-controlled datasets. This is because these in-the-wild datasets have considerable advantages in the sample size and diversity, making the model learn robust information. To our surprise, compared with DAN, our ESSRN also made greater progress on the experiments whose source domain is RAF-DB or FER2013. One possible reason for this phenomenon is that, compared with the lab-controlled datasets, the datasets in the wild are prone to generate outlier samples caused by subjective annotation, image occlusion, and other uncontrollable reasons. However, our model is quite sensitive to outlier samples in the training set and is good at reducing or even eliminating the impact of outlier samples on the cross-dataset FER.

Finally, compared with AGRA [38], one of the most currently available comprehensive and state-of-the-art methods, our ESSRN fails to achieve the best results in the *R→F*, *J→C*, and *F→C* experiments. We speculated that this is due to AGRA paying extra attention to the local information of the face based on the landmarks, and the common facial features can provide more detailed information for cross-dataset facial expression recognition. In addition, among these experiments, our ESSRN performed poorly in the *J→C* experiment, which we believe is due to the small sample size and almost no outlier sample in the source domain, which hinders the performance of our ESSRN.

In addition, we also visualize the OPC of samples in RAF-DB to investigate the effectiveness of our ESSRN. The experimental results reported in Figure 4 show that our ESSRN gives a lower OPC to facial images with low quality, huge differences in personal attributes (e.g., age and ethnicity), inconsistent annotation, and serious occlusion while giving a higher OPC to facial images with clarity, no objection, and no occlusion. The experimental results indicate that the proposed ESSRN can effectively perceive outlier samples in the training set and reduce the proportion of these outlier samples in the model training.

### 5.2. Evaluation of Trade-Off Parameters

In this part, we conduct the sensitivity analysis of trade-off parameters on the experiment, which treats RAF-DB as the source domain and CK+ as the target domain. In the first experiment, we investigate the impact of λ, which represents the trade-off between $L_{OP}$ and $L_{MMD}$. We set γ=0.6 and $\lambda =\left[0.0001,0.001,0.01,0.1,1\right]$ to conduct different cross-dataset FER experiments. The verification accuracies of these models are shown on the left side of Figure 5. It can be observed that properly choosing the value of the trade-off parameter λ can improve the recognition accuracy. In the second experiment, we explore the impact of γ, which indicates the trade-off between $L_{OP}$ and $L_{OR}$. We set λ=0.01 and $\gamma =\left[0,0.2,0.4,0.6,0.8,1.0\right]$ to learn different models. The verification accuracies of these models are shown in the right side of Figure 5. It can be observed that the accuracies of our ESSRN remain largely stable across a wide range of the trade-off parameter γ, which demonstrates that our ESSRN is less sensitive to the choice of trade-off parameters λ and γ.

### 5.3. Ablation Study

An ablation study is conducted to demonstrate the effect of each module of our ESSRN. Four datasets are selected as the source domain, respectively, and one of the datasets is randomly selected as the target domain to conduct the ablation experiments. Some conclusions can be observed from experimental results shown in Table 4. First of all, compared to the baseline in the first row, which ResNet-18 conducts, each module in ESSRN improves the performance of cross-dataset expression recognition. Then, the second row in Table 4 reveals that ESSRN has limited improvement when the model only has the outlier perception module. This is because the outlier perception module can work properly only with the constraint of the OPC revision module. Otherwise, the OPC generated by the outlier perception module will become meaningless and become closer to one in value after many rounds of training. Thirdly, the added OPC revision module improves the performance of ESSRN, which is consistent with the previous analysis, while the feature transfer module can further enhance the model’s performance. Finally, it can be easily found that the biggest improvements are all achieved by the feature transfer module in four experiments, which increases by 2.82% in *R→J*, 8.74% in *J→C*, 4.23% in *C→J* and 3.04% in *F→R*, respectively. This indicates the great advantages of applying our well-designed UDA module.

## 6. Conclusions

In this paper, we proposed the enhanced sample self-revised network (ESSRN) with novel outliers handling mechanism to solve the impact of outlier samples in cross-dataset FER tasks, which hinders the UDA models from learning the discriminative facial expression features. Our ESSRN can perceive the outlier level of the source data with the help of the outlier perception and OPC revision modules. Via tagging different OPC to different samples in the source domain, the proposed ESSRN is able to weaken or strengthen the contribution of outliers or other samples to the model training, respectively. Furthermore, the feature transfer module first constrains the distribution distance of high-coefficient group samples and low-coefficient group samples in the source domain to revise outlier samples in the source domain at the feature level. Then, the feature transfer module aligns the marginal and class-conditional distribution of the source and target domains with obtaining more robust domain invariant features. Extensive cross-dataset FER experiments between CK+, JAFFE, RAF-DB and FER2013 datasets were conducted to evaluate the performance of our ESSRN. Experimental results indicated that the proposed outliers handling mechanism can reduce the negative impact of outlier samples on cross-dataset FER effectively and our ESSRN achieves a better performance than the traditional deep UDA methods and most state-of-the-art cross-dataset FER methods.

## Figures and Tables

**Figure 1 entropy-24-01475-f001:**
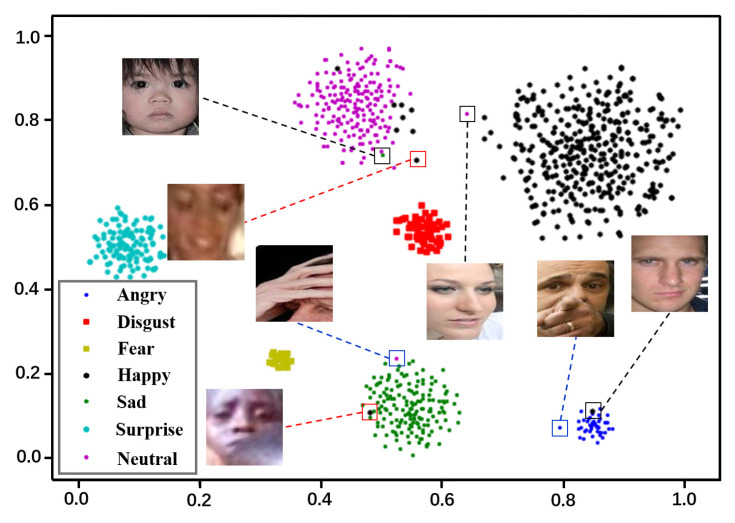
Visualization of outlier samples in RAF-DB dataset by t-SNE. We randomly select 1000 samples from the RAF-DB dataset and extract their 512-dimensional features in the last layer of ResNet-18. It is clear to see that several outlier samples indeed exist in the dataset, where the RED, BLUE, and BLACK Rectangle Marquees highlight the samples with low quality, subjective annotation, and severe occlusion, respectively.

**Figure 2 entropy-24-01475-f002:**
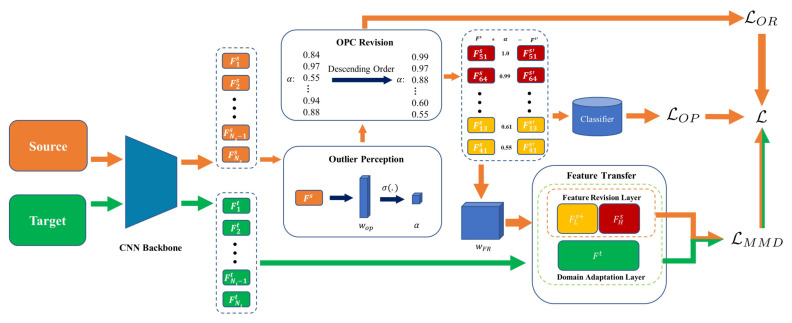
The illustration of ESSRN: orange and green rectangular blocks represent the source and target domains. The red and yellow blocks indicate the features from high-coefficient group and low-coefficient group in the source domain. The orange and green flows represent processes for source and target domains, respectively.

**Figure 3 entropy-24-01475-f003:**
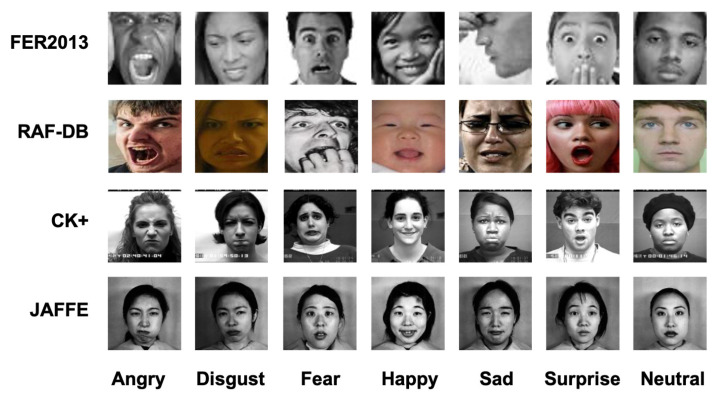
Typical sample images under seven categories from FER2013, RAF-DB, CK+ and JAFFE datasets presented.

**Figure 4 entropy-24-01475-f004:**
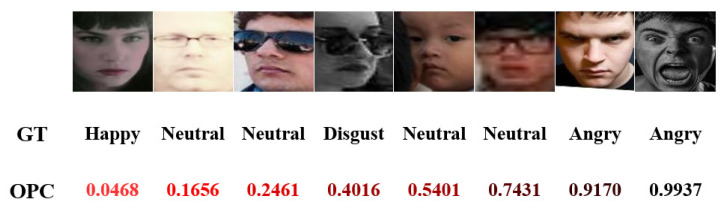
Visualization of OPC in RAF-DB learned by ESSRN: GT represents the ground truth of the corresponding samples.

**Figure 5 entropy-24-01475-f005:**
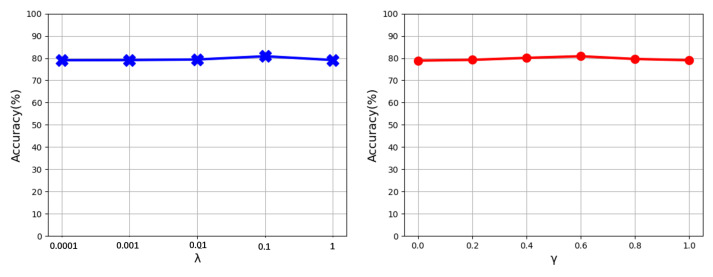
Evaluation of hyper parameter λ and γ on the experiment which treats RAF-DB as the source domain and CK+ as the target domain.

**Table 1 entropy-24-01475-t001:** Sample statistics of RAF-DB, JAFFE, CK+ and FER2013 for experiments.

Dataset	Angry	Disgust	Fear	Happy	Sad	Surprise	Neutral	Total
FER2013	4953	547	5121	8989	6077	4002	6198	35,887
RAF-DB	867	877	355	5957	2460	1619	3204	15,339
CK+	135	177	75	207	84	249	309	1236
JAFFE	30	29	32	31	31	30	30	213

**Table 2 entropy-24-01475-t002:** Results of cross-dataset FER experiments between RAF-DB, JAFFE, CK+ and FER2013. The method with the highest accuracy in each experiment is indicated in bold.

Experiments	DANN [35]	DAN [36]	DETN [16]	ICID [37]	ECAN [12]	AGRA [38]	SSRN [15]	ESSRN
*R→J*	63.85	62.44	52.11	48.83	52.11	61.03	**67.61**	63.85
*R→C*	74.76	75.49	64.19	67.44	66.51	77.52	**80.99**	80.83
*R→F*	45.94	50.05	42.01	53.00	50.76	**54.94**	50.83	50.98
*J→R*	36.52	38.84	26.77	16.27	27.13	39.71	38.84	**40.11**
*J→C*	48.46	65.94	31.01	24.03	40.31	**82.17**	63.03	68.85
*J→F*	27.59	25.14	27.03	18.54	27.06	28.44	25.05	**28.54**
*C→R*	36.97	43.68	23.26	19.73	26.04	35.44	**45.67**	44.52
*C→J*	49.30	56.81	16.54	27.23	24.24	49.30	56.81	**57.75**
*C→F*	31.14	33.08	18.74	23.77	19.33	30.89	29.98	**34.54**
*F→R*	59.81	63.60	57.14	60.71	59.35	65.12	62.08	**66.59**
*F→J*	52.11	54.46	33.65	38.50	37.86	48.16	55.40	**65.26**
*F→C*	71.52	76.21	72.09	68.22	65.89	**80.95**	69.82	79.53
Average	49.83	53.81	38.71	38.86	41.38	54.47	53.84	**56.78**

**Table 3 entropy-24-01475-t003:** Average performance of experiments with the same source domains. The third column and the fourth column respectively represent the average accuracy of DAN and our ESSRN with the same source domain, and the last column indicates the increment of our ESSRN compared to DAN.

Source Set	Target Set	DAN [36]	ESSRN	Increment
*FER2013*	*Average*	64.76	70.46	5.70
*RAF-DB*	*Average*	62.66	65.22	2.56
*JAFFE*	*Average*	43.31	45.83	2.52
*CK+*	*Average*	44.52	45.60	1.08

**Table 4 entropy-24-01475-t004:** Ablation study in ESSRN.

Method	*R*→*J*(%)	*J*→*C*(%)	*C*→*J*(%)	*F*→*R*(%)
Baseline	59.15	45.79	49.01	62.60
$L_{OP}$(ESSRN)	60.09	52.02	50.23	62.78
$L_{OP}+L_{OR}$(ESSRN)	61.03	60.11	53.52	63.55
$L_{OP}+L_{OR}+L_{MMD}$(ESSRN)	63.85	68.85	57.75	66.59

## Data Availability

Not applicable.

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
