# Peer review of "Enhanced Sample Self-Revised Network for Cross-Dataset Facial Expression Recognition"

_entropy, 2022, doi:10.3390/e24101475_

Round 1

Reviewer 1 Report

Comments to the Author

This paper proposes the Enhanced Sample Self-Revised Network (ESSRN) based on authors previous conference work of SSRN [17]. The structure of the paper is clear; however, there are some major issues that need to be addressed:

1. Algorithm 1: Update Line 2 with clear statement. Which line computes ESSRN?

2. The results are not very interesting and require more comparison with state of the art, for example, Huilin et. al, “Facial expression recognition based on deep learning, ” Computer Methods and Programs in Biomedicine.

3. The paper must be checked for grammar and typos.

Reviewer 2 Report

The paper written is on a very interesting subject domain namely, “cross-dataset facial expression recognition” emphasizing on outliers in large scale facial image datasets. However, certain suggestions are given to the authors for consideration.
1.    Some reflection on the results should be made in the abstract.
2.    Sentences are awfully long in length and readers usually strides away from the actual meaning during reading.
3.    Overview of ESSRN should be reconsidered for writing. The topic contains various technical and grammatical anomalies.
4.    What is the possibility of learning incorrect facial expression features by the model. Your claim should be supported by a strongly cited work as well.
5.    The work makes use of various approaches as a pipeline. In the presence of the work being used by the study, please explain the novelty of your work and justify it with the support of your results.
6.    What is the significance of this work in comparison to deep models used for similar studies. Please include comparative results with such studies.

7. Majority of the references are outdated. Cite recent works for better readability and visibility of the manuscript.

Round 2

Reviewer 1 Report

Thank you, all my previous comments were addressed.